# Impact of Vitiligo on Life Quality of Patients: Assessment of Currently Available Tools

**DOI:** 10.3390/ijerph192214943

**Published:** 2022-11-13

**Authors:** Ting-Ting Yang, Chien-Hung Lee, Cheng-Che E. Lan

**Affiliations:** 1Department of Dermatology, Kaohsiung Medical University Hospital, Kaohsiung Medical University, Kaohsiung 807, Taiwan; 2Department of Dermatology, Pingtung Hospital, Ministry of Health and Welfare, Pingtung 900, Taiwan; 3Department of Public Health, College of Health Sciences, Kaohsiung Medical University, Kaohsiung 807, Taiwan; 4Research Center for Environmental Medicine, Kaohsiung Medical University, Kaohsiung 807, Taiwan; 5Department of Dermatology, College of Medicine, Kaohsiung Medical University, Kaohsiung 807, Taiwan

**Keywords:** mental health, patient health questionnaire, pigmentation disorders, psychologic stress, quality of life, vitiligo

## Abstract

Background: How different tools for evaluating health-related quality of life (QoL) reflect the burden of vitiligo has rarely been compared. In this study, we aim to evaluate the impact of vitiligo on QoL using currently available tools. Methods: A single-center, cross-sectional study of vitiligo patients was performed. QoL was evaluated using the dermatology life quality index (DLQI), short form-36 (SF-36), and willingness to pay (WTP). As the original DLQI cutoff score (>10) indicating aginificantly impated QoL was found to underestimate QoL, receiver operating characteristic (ROC) curve was use to determine a new cutoff point discriminating vitiligo patients with positive mental stress (SF-36 mental health ≤ 52). Results: Of the 143 patients enrolled, 24.5% were identified having significant mental stress by SF-36 mental health domain score ≤ 52 while there were only 13.3% patients with significantly hampered QoL indicated by DLQI >10. Using ROC analysis, DLQI ≥ 7 was found to be a more appropriate DLQI cutoff value for identifying mental stress in vitiligo patients. Additionally, the median WTP for treating vitiligo was comparable to other inflammatory skin diseases. Conclusions: There exist discrepancies in the outcomes identifying significant disease burden of vitiligo using DLQI and SF-36, making the current DLQI cutoff score (>10) for identifying significantly affected QoL inappropriate for these patients.

## 1. Introduction

Pigmentary disorders are identified when the skin tone deviates from the expected normal skin color. Despite significantly impacting the life quality of those affected, these disorders are often neglected due to their lack of association with direct life threats [1]. One prototypical pigmentary disorder is vitiligo, an acquired depigmentary disorder with significant genetic influence affecting approximately 1% of the world population [2]. Although vitiligo seldom causes symptoms and is usually not life threatening, it negatively affects patient quality of life (QoL), especially the mental health of patients [3,4,5].

Currently, the impact of diseases on QoL is often evaluated via validated questionnaires. Short form 36 (SF-36), a health survey that evaluates the generic health related well-being comprised of eight specific domains, is widely used to assess the health related QoL in various diseases. Dermatology life quality index (DLQI) has been established especially for skin diseases and is believed to be the most appropriate tool for assessing burden of skin diseases on patients [6,7]. It provides a basis for comparing QoL among different dermatologic diseases and is the recently recommended instrument for evaluating QoL of vitiligo patients by the EADV Task Force on QoL and Patient Oriented Outcomes [8]. Other approaches to evaluate the impact of disease include evaluating willingness to pay (WTP) for disease treatment. More specifically, WTP is commonly used in health-economics studies to estimate the monetary value of health benefit and patient preference by asking the maximum amount a person is willing to spend for disease treatment. Previously, the WTP for various dermatologic diseases, including psoriasis and atopic dermatitis, has been explored [9,10,11].

As health-related QoL measurements are essential for monitoring patient progress and influence of therapy, understanding how the outcomes of these tools differ is important while managing patients with skin diseases. The above tools have been chosen in the present study since they have been used for evaluating different dermatologic conditions and are able to provide a basis for comparison of the impact of different diseases, including non-dermatologic diseases. However, different tools may reflect disease burden differently. For example, DLQI comprises symptoms and signs into a composite score, which may potentially underestimate the impact of diseases without obvious symptoms, such as vitiligo. On the other hand, SF-36 is a more generalized questionnaire which consists of different domains that specific aspects of the impact of a disease can be evaluated independently. Different from the above tools, WTP is a form of contingent valuation estimating values in monetary terms and may capture aspects of disease burden not included in standardized questionnaires. The objective of this study is to compare the outcomes of these different tools commonly used to assess impacts of vitiligo on affected patients. Additionally, since DLQI is frequently used to evaluate the impact of dermatologic conditions on affected patients, the current study also aimed to develop a more suitable cutoff score indicating severe impact on QoL as the current critierion (>10)was found to underestimate the impact of vitiligo on patients’ QoL.

## 2. Materials and Methods

### 2.1. Subjects

A single-center, cross-sectional study on consecutive patients who were ≥ 20 years of age with a clinical diagnosis vitiligo seeking treatment from April 2019 to March 2020 at the Department of Dermatology of Kaohsiung Medical University Hospital was designed. Patients with multiple dermatologic disorders (also having dermatologic diseases other than vitiligo) were excluded but those with concomitant non-dermatologic diseases were included. All patients enrolled gave written consent to participate. This study was approved by the hospital institutional review board (KMUHIRB-E(I)-20190114).

### 2.2. Instruments for Outcome Measurements

Participants completed a standardized questionnaire including questions regarding demographic characteristics (age and sex), disease duration, monthly income (categorized into six groups: below 26,000 new Taiwan dollars (NTD), 26,001 to 39,000 NTD, 39,001 to 50,000 NTD, 50,001 to 100,000 NTD, and over 100,000 NTD), and WTP for vitiligo treatment. The cutoff values for each group in the monthly income question were determined according to the 2018 Taiwan Employees’ Earning Survey, where wages less than 26,000 NTD per month indicated low income, above 100,000 NTD per month indicated high income, 39,000 NTD per month represented the median, and 50,000 represented the mean monthly wage [12]. Willingness to pay was evaluated by a single closed question asking participants the proportion of monthly income the participant was willing to spend on curative treatment. The possible answers were <10%, 10–19%, 20–29%, 30–39%, and ≥40% of income per month [13,14]. Higher WTP indicates increased disease burden [9,10]. All questions were written in traditional Chinese. For participants without adequate language proficiency, an interviewer was provided to read the questions to the participant in either Mandarin or Taiwanese.

### 2.3. Quality of Life

Health related quality of life was measured by the traditional Chinese version of DLQI translated from the original English questionnaire and validated traditional Chinese version of SF-36 health survey [15]. DLQI is a widely used questionnaire in dermatology for evaluating the impact of skin diseases on QoL [6,7]. It contains ten questions measuring six dimensions of life quality including symptoms (pain, itch, sore, and stining) and feelings (embarrassed or self conscious), daily activities (shopping, gardening, choosing clothing), social or leisure activities (including doing sports), work and school, personal relationships (including sexual activities), and treatment. The scores of each question are added to obtain a total score ranging from 0 to 30 points. The full questionnaire is provided in supplement 1 (downloaded from “https://www.cardiff.ac.uk/medicine/resources/quality-of-life-questionnaires/dermatology-life-quality-index” (accessed on 10 November 2022)). More specifically, DLQI scores greater than 10 indicates QoL impairment with very large effect on patient’s life [6,7]. The SF-36 health survey questionnaire is a validated and reliable instrument evaluating health-related QoL [15]. It assesses health quality on eight domains: physical function (PF), role limitation related to physical problems (RP), bodily pain (BP), general health (GH), social functioning (SF), vitality (VT), role limitation related to emotional problems (RE), and mental health (MH). Scoring for each domain ranges from a scale of 0–100. Higher scores indicate better QoL. The above subscores are further summarized into physical component score (PCS) and mental component score (MCS) using the method described in Taft et al. [16]. Both PCS and MCS are standardized scores with a mean of 50 and standard deviation of 10 for the Taiwanese norm population. Hence, scores above 50 indicate QoL of the better than the general population and vice versa. Mental health (MH) domain scores ≤ 52 is used to indicate an emotional limitation as described in the original SF-36 Health Survey Manual & interpretation guide by Ware et al. [17]. Therefore, MH ≤ 52 may be considered as a threshold indicating significant hampered QoL using SF-36 assessment.

### 2.4. Disease Severity and Location of Involvement

Photography of the lesions was taken at time when patients were enrolled in the study. For evaluation of disease severity, the vitiligo extent score (VES) was used [18]. Higher VES indicated more extensive body surface area involvement. The locations of the vitiligo lesions were documented for every patient. Visible lesions at the head, neck, and hands were considered as exposed lesions and other lesions were considered as non-exposed.

### 2.5. Statistical Analysis

All statistical computations were performed by SPSS software (Chicago, IL, USA) version 21.0. Data normality were checked using the Shapiro–Wilk test for all variables before analysis. For non-normally distributed continuous and ordinal data, Wilcoxon rank-sum test was used to analyze the difference between groups. Spearman’s rank test was used to determine the level of association between variables. *p*-value equal or less than 0.05 was considered significant. Participants unwilling to answer questions regarding monthly income were excluded from WTP analysis. Because DLQI score was found to be significantly associated with SF-36-based MH, we used receiver operating characteristic (ROC) analysis, an established method for determining the optimal threshold for a binary classifier system, to determine an optimal cutoff point that discriminates the positive mental stress (defined as the SF-36 MH domain score ≤ 52) among vitiligo patients. The SF-36-based MH was our primary binary outcome and the DLQI score was our explanatory variable. In a priori sample size estimation given two-sided type I error of 5% and standard deviation of 4.5 of DLQI score, 143 patients were calculated to provide 80% statistical power for detecting an effect size of 1.5 in DLQI scores between patients with MH ≤ 52 and MH > 52. The sensitivity (true positive rate) and 1-specificity (false positive rate) of each potential threshold is plotted on the ROC curve. The best cutoff point for the identification of positive mental stress were determined by maximizing the Youden’s index [19].

## 3. Results

### 3.1. Patient Demographics

A total of 143 consecutive vitiligo patients were enrolled. All patients were of East Asian descent with Fitzpatrick skin type III or IV. The characteristics of all participants are summarized in Table 1. Of all patients enrolled, the monthly income and WTP was recorded in 81 participants. The median income of the patients enrolled was within the group 39,001 NTD–50,000 NTD, which is between the median and mean monthly income of Taiwanese workforce [12].

### 3.2. Quality of Life

The mean (SD) DLQI score was 5.32 (4.67) and 19 patients (13.3%) had DLQI scores greater than 10, indicating significantly hampered QoL. The mean (SD) PCS and MCS scores were 51.72 (6.02) and 47.15 (7.99), respectively, and SF-36 scores for each domain and component scores are summarized in Table 2. As aforementioned, MH domain score ≤ 52 indicates significant impairment. Approximately a quarter of the participants (*n* = 35, 24.5%) in this study had MH domain scores less than or equal to 52. After removing patients with other comorbidities (*n* = 117), the proportion of patients identified as having severely affected disease by DLQI and MH score were 13.7% (16/117) and 25.6% (30/117), respectively. The means of QoL scores, including those derived from DLQI and SF-36, were not significantly different between female and male participants. Additionally, the mean scores derived from both instruments showed no significant difference between patients with exposed and non-exposed lesions. Subsequent analysis of the factors associated with worse QoL showed a significant positive correlation between disease severity and DLQI (Spearman’s Rho = 0.25, *p* = 0.003) and a nearly significant negative association between severity and MCS (Spearman’s Rho = −0.165, *p* = 0.051) (Table 3). Within the eight domains of SF-36 health survey, severity was found to significantly and negatively correlated with SF (Spearman’s Rho = –0.232, *p* = 0.006). It is intriguing to note that except for PF and BP, DLQI scores showed significant negative correlation with 6 out 8 domains within SF-36. The DLQI scores showed significant negative correlation with MCS (Spearman’s Rho = −0.488, *p* < 0.001) but not with PCS (Spearman’s Rho = −0.007, *p* = 0.932). Therefore, for vitiligo patients, DLQI scores reflect the mental stress but not physical impairment of the affected individuals.

### 3.3. Willingness to Pay

Since a proportion of patients refused to reveal their personal income due to privacy reasons, only 81 patients were included for analysis. The median WTP for vitiligo treatment was 10–19% of monthly income. Interestingly, 11% of patients were willing to pay more than 40% of their monthly income for complete cure of vitiligo. Similar to DLQI and SF-36, those who have lesions on exposed areas did not have a higher median WTP (*p* = 0.383). Both severity (*p* = 0.579) and disease duration (*p* = 0.579) did not show a significant correlation with WTP. Additionally, no difference was found between genders (*p* = 0.698) in regard to median WTP. Finally, the level of income did not show a significant correlation with WTP.

### 3.4. Deriving New Cutoff Value for DLQI

Figure 1 presents the discriminatory ability of DLQI score in identification of positive mental stress defined as MH ≤ 52 among vitiligo patients. The DLQI score revealed a significant capability in determining positive mental stress, with a 0.749 (95% confidence interval: 0.655–0.844) of the area under ROC curve (*p* < 0.05). The best cutoff value of DLQI was 7 (DLQI ≥ 7), at which the Youden index exhibited a maximum sensitivity (62.9%) and specificity (75.0%) in determining positive mental stress.

## 4. Discussion

Understanding the disease burden is crucial in disease management. For vitiligo, the QoL of patients can be evaluated using generalized health related QoL questionnaires such as SF-36, questionnaire designed for skin diseases such as DLQI, or specific assessment tools, including vitiligo quality of life index (VitiQoL), vitiligo impact scale-22 (VIS-22) and vitiligo impact patient scale (VIPs) [20,21,22]. However, validated translated versions of the aforementioned specific questionnaires are not readily available and do not allow for comparison with the QoL of the general population or other diseases. Besides, the outcomes of these disease specific tools may be subjected to cultural influences [8]. In this study, we examined the impact of vitiligo via DLQI, SF-36, and WTP assessments.

DLQI is a standardized QoL questionnaire established specifically for skin diseases. The mean DLQI score of vitiligo patients in the current literature ranges from 4 to 10.67 [3]. This variation is likely due to ethnic and cultural difference, and the mean DLQI (5.32) in the present study also fell within this interval. Among studies from East Asian countries, where Taiwan shares similar ethnic and cultural background, the DLQI scores of vitiligo patients rages from 5.9 (Japan) to 5.83–8.41 (China) [23,24,25]. Compared with the QoL of other dermatologic diseases characterized with prominent symptoms including itch and pain, the DLQI of patients in this study was lower than psoriasis (DLQI = 8.34) [9] and hand eczema (DLQI = 7.31) [10], suggesting better QoL of vitiligo compared to skin disorders with prominent symptoms according to DLQI scores. Similarly, vitiligo patients were reported to have lower DLQI scores in studies directly comparing the QoL of vitiligo and other dermatologic diseases [26,27,28]. However, do these reports accurately reflect the true impact of vitiligo on QoL of patients affected with this depigmentary condition?

Compared with the SF-36 scores derived from Taiwanese population [29], vitiligo patients had lower mental QoL outcomes but comparable physical QoL. Participants in this study had lower mean MCS and lower SF-36 scores in three of the four mental domains including VT, SF, and MH domains as compared to the general Taiwanese population. As described in the original SF-36 Health Survey Manual & interpretation guide, a cutoff value of MH ≤52 indicates severe emotional limitation based on the association of MH scores and other psychiatric disorders [17]. By using this cutoff value, a substantial proportion of vitiligo patients in the present study (24.5%) were considered to harbor a high risk for developing psychiatric disorders. These findings are consistent with previous reports showing that vitiligo patients have increased psychological stress and a higher prevalence of psychiatric comorbidities [4,5,30,31]. Intriguingly, recent studies showed that approximately 30% of vitiligo patients have moderate to severe depression and anxiety, and the prevalence of depression is 25.3% among vitiligo patients [32,33]. Taken together, these results suggest that cutoff value for MH less than 52 may provide a good estimation for identifying vitiligo patients who are at risk for having emotional distress, and approximately a quarter of vitiligo patients are at high risks for developing psychiatric problems.

Although DLQI has significant negative correlation with MCS and 6 out 8 domains within SF-36, the proportion of significantly affected patients, as identified by DLQI (DLQI > 10) and SF-36 (MH ≤ 52), showed significant differences (13 vs. 24%, respectively). Since the DLQI questionnaire incorporates questions focusing on the impact of certain symptoms, including pain or itch [6], when using DLQI for evaluation for skin conditions with obvious clinical signs but limited symptoms, the cutoff value indicating significant impairment in these diseases must be modified. Moreover, it should be noted that in the original study reported by Hongbo et al. [6], who first advocated using DLQI > 10 as the cutoff score indicating significantly impaired QoL, patients with pigmentary disorders were under-represented, which further indicated that the current established cutoff value is inappropriate for determining the significance of QoL impairment for skin conditions mainly affecting the pigmentary status of the skin such as vitiligo. To address this issue, we conducted ROC analysis and revealed that DLQI ≥ 7 is a more appropriate cutoff value for identifying severely affected vitiligo patients. Since DLQI is frequently used to compare the impact different skin diseases on the QoL of affected patients, this new cutoff score gives a reference for future researchers and clinicians to identify vitiligo patients with severely affected QoL.

In addition to health related QoL questionnaires, evaluation of WTP renders further insights into the disease burden of vitiligo. While compared with other dermatologic diseases, the median WTP for vitiligo was comparable to psoriasis (median: 9.8% of monthly income) [11], supporting the notion that the disease burden of vitiligo, a pigmentary disease, is at least as severe as psoriasis, an inflammatory skin disorder. Similarly, a recent study directly comparing the WTP and DLQI scores between vitiligo and other dermatologic disorders also reported that vitiligo patients had the highest median WTP for disease cure but lowest median DLQI score [34]. Since WTP assessment evaluates the disease burden as a whole and not in fragmented domains, this instrument may capture aspects of QoL not included DLQI [11,34]. Taken together, this result suggests that for skin diseases like pigmentary disorders that harbor obvious clinical signs but limited symptoms, DLQI scores may significantly underestimate the burden of disease on the affected patients. Therefore, besides lowering the DLQI cutoff score indicating severe disease for pigmentary disorders, adding questions regarding WTP may greatly enhance the estimation accuracy of QoL in these patients.

The present study has several limitations. First, only those who sought medical treatment were included in the present study and may potentially result in certain selection bias. However, since the objective of this study is to compare the outcomes of different tools used to assess impacts of vitiligo on affected patients and all the participants responded to different assessment tools, the potential selection bias has limited effect on our analyses. Second, since income is a relatively sensitive issue, not all patients completed questions regarding WTP, which may also induce selection bias. Nonetheless, there exists no evidence that patients refusing to reveal income information may belong to a certain income group. Hence, the results may not be skewed towards a certain income level even with high refusal rates since the income of those refusing to answer WTP questions are more likely to be randomly distributed. Thirdly, the cutoff value (MH ≤ 52) indicating severe emotional limitation was established on the US population and may be susceptible to cultural influences. However, the population norms for mean MH scores were not significantly different between the US and Taiwanese populations (74.74 vs. 73.01) [17,29]. Moreover, the SF-36 questionnaire does not contain questions that may be influenced by cultural differences, making its outcome less likely to be biased by cultural differences. Lastly, the impacts of vitiligo can be influenced by the natural skin color of patients. It should be reminded that all patients in the present study have Fitzpatrick skin type III or IV and the results may only be applicable to those of similar skin color.

## 5. Conclusions

This study highlighted several important messages. First of all, vitiligo significantly hampers the QoL of those affected. Secondly, DLQI is a good instrument documenting the impact of skin diseases on QoL as DLQI scores show a significant correlation with disease severity of vitiligo. Thirdly, the different components of SF-36 served to give specific insights on different aspect of disease, and the severity of vitiligo correlated most significantly with the SF of the patient. Last and most importantly, there exist discrepancies identifying significant disease burden of vitiligo as identified by different tools. The cutoff score (>10) indicating significant QoL impairment should be modified for skin conditions with limited symptoms to truly reflect the burden of disease, and we propose that modifying the DLQI cutoff value to ≥7 is more appropriate to identify severely affected patients with vitiligo. This new cutoff score offers a reference point for the future researchers to identify vitiligo patients with severely affected QoL.

## Figures and Tables

**Figure 1 ijerph-19-14943-f001:**
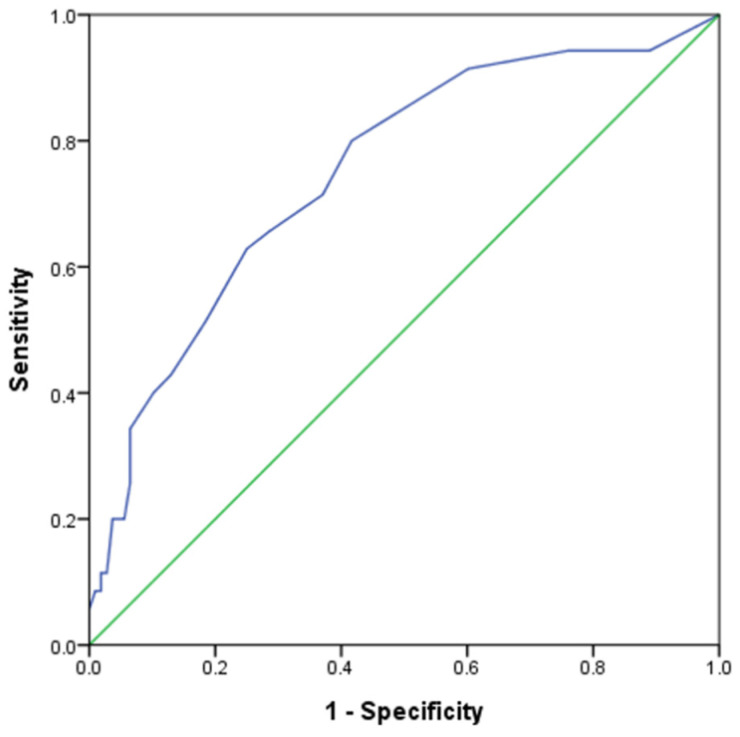
The receiver operating characteristic (ROC) curve associated with the discrimination of dermatology life quality index (DLQI) score on positive SF-36-based mental stress. Note: Positive SF-36-based mental stress was defined as the score of SF-36 mental health domain ≤52. The area under ROC curve was 0.749 (95% CI: 0.655–0.844), *p* < 0.05. The cutoff point of DLQI was 7.

**Table 1 ijerph-19-14943-t001:** Patient characteristics.

Characteristics	*n* (%)
Sex (female), *n* = 143	86 (60.1)
Age, mean (SD), *n* = 143	44.9 (14.8)
Severity, *n* = 143	
VES, median (IQR) Mean (SD)	0.56 (0.24–1.35) 1.26 (2.15)
Location, *n* = 143	
Exposed	123 (86)
Non-exposed	20 (14)
Disease duration, *n* = 143	
<6 months	18 (12.6)
6 months–1 year	15 (10.5)
1 year–3 years	33 (23.1)
3 years–5 years	15 (10.5)
>5 years	62 (43.4)
Concomitant diseases, *n* = 143	
None	117 (81.8)
Malignancy	2 (1.4)
Thyroid disorders	7 (4.9)
Diabetes mellitus	2 (1.4)
Hypertension	6 (4.2)
Sjogren’s syndrome	1 (0.7)
Coronary artery disease	3 (2.1)
* Other diseases	8 (5.6)
Monthly income (NTD), *n* = 81	
<26,000	23 (28.4)
26,000–39,000	16 (19.8)
39,000–50,000	22 (27.2)
50,000–100,000	14 (17.3)
>100,000	6 (7.4)

Legend: IQR, interquartile range; SD, standard deviation; NTD, new Taiwan dollars. * Other diseases include osteoarthritis (*n* = 1), adenomyosis (*n* = 1), gastroesophageal reflux disease (*n* = 2), cardiac arrhythmia (*n* = 2), and benign prostate hyperplasia (*n* = 2).

**Table 2 ijerph-19-14943-t002:** Quality of life outcomes.

	Mean (SD)	Median (IQR)
DLQI, *n* = 143	5.32 (4.67)	4 (2–8)
SF-36, *n* = 143		
Physical functioning (PF)	93.6 (10.18)	100 (90–100)
Role-physical (RP)	91.61 (22.58)	100 (100–100)
Bodily pain (BP)	88.86 (16.38)	100 (82–100)
General health (GH)	61.97 (18.61)	62 (48.5–72)
Vitality (VT)	62.21 (16.21)	60 (50–70)
Social function (SF)	84.79 (15.78)	87.5 (75–100)
Role-emotional (RE)	84.62 (30.58)	100 (100–100)
Mental health (MH)	64.62 (16.30)	64 (56–76)
	*n* (%)	
Willingness to pay, *n* = 81		
<10%	29 (35.8)	
10–19%	32 (39.5)	
20–29%	8 (9.9)	
30–39%	3 (3.7)	
≥40%	9 (11.1)	

Legend: DLQI, dermatology life quality index; SF-36, Short form 36; SD, standard deviation; IQR, interquartile range.

**Table 3 ijerph-19-14943-t003:** Factors associated with QoL outcomes.

	DLQI		PF		RP		BP		GH	
	Rho	*p*-Value	Rho	*p*-Value	Rho	*p*-Value	Rho	*p*-Value	Rho	*p*-Value
DLQI	-	-	−0.079	0.351	−0.173	**0.039**	−0.134	0.112	−0.280	**0.001**
Severity	0.25	**0.003**	−0.016	0.854	−0.105	0.217	−0.094	0.267	0.091	0.282
	**VT**		**SF**		**RE**		**MH**		**WTP**	
	**Rho**	* **p** * **-value**	**Rho**	* **p** * **-value**	**Rho**	* **p** * **-value**	**Rho**	* **p** * **-value**	**Rho**	* **p** * **-value**
DLQI	−0.331	**<0.001**	−0.284	**0.001**	−0.289	**<0.001**	−0.466	**<0.001**	-	-
Severity	0.010	0.903	−0.232	**0.006**	−0.048	0.569	−0.133	0.115	0.055	0.623

Legends: DLQI, dermatology life quality index; PF, physical function; RP, role limitation related to physical problems; BP, bodily pain; GH, general health; SF, social functioning; VT, vitality; RE, role limitation related to emotional problems; MH, mental health. Bold numbers indicate significant results (*p*-value < 0.05).

## Data Availability

Datasets used in the present study have been deposited in Mendeley Data (“https://doi.org/10.17632/9tkgbfghyw.1” (accessed on 10 November 2022)).

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
