# Peer review of "Impact of Vitiligo on Life Quality of Patients: Assessment of Currently Available Tools"

_ijerph, 2022, doi:10.3390/ijerph192214943_

Round 1
Reviewer 1 Report (Previous Reviewer 1)
All the points raised have been addressed reasonably by the authors. The revised manuscript looks fine. The manuscript may be accepted.Author Response
We would like to thank the reviewer for his/her kind comments.
Reviewer 2 Report (New Reviewer)
As a cross-sectional study, the article is of good quality. However, the background and discussion sections must be improved. I list my concerns below.
Introduction.
The necessitary of the study should be further elaborated. A more suitable cutoff score will not address the problem of the DLQI questionnaire underestimating the impact of disease.
If WTP is another outcome, it should be listed in the objective. More background information on WTP should be provided.
Methods.
Inclusion and exclusion criteria should be described more clearly. For example, how did you deal with the patients with non-dermatologic diseases but having skin complications (e.g., diabetic foot)?
Line 79. It is a common confusion that higher WTP indicates worse QoL. The sentence should be corrected. The authors should also provide more evidence here to support themselves.
Have the traditional Chinese version of the DLQI and SF-36 questionnaires been validated? More information should be provided.
Line 117. How was "missing" defined? If a patient missed a question or a characteristic data, will he be excluded?
The approval number of ethic review should be provided.
Discussion.
Psoriasis also causes discolored patches of skin, and vitiligo can be autoimmune and inflammatory. The comparison of pigmented and inflammatory diseases is inappropriate.
Line 222. There are some previous observational studies as authors stated, and the word "real-world" made it confusing here.
Line 276. Should explain why this inference is made.
Importantly, the authors did not state the contributions of their study to the field, nor the positive influence on future clinical practice.
Round 2
Reviewer 2 Report (New Reviewer)
The authors have improved the manuscript a lot. Some improvements are still needed.
Abstract
Under methods, the analyses should be reported, so the results can be understood without having to refer to the main text.
Introduction
The advantages of the Dermatology life quality index (DLQI) over other tools is still not clear enough. More details must be provided.
Methods
Who did document the locations of the vitiligo lesions?
As a not very commonly used QoL tool, the full content of the Dermatology life quality index (DLQI) should be provided in the supplementary materials.
Why did the authors use MH domain as an indicator of emotional limitation, instead of MCS, since MCS has already been standardized? The same question applies to using MH ≤ 52 as a threshold indicating significant hampered QoL instead of MCS. I do not see the point here.
Discussion
Line 205. In my opinion, DLQI should be classified as a disease-specific assessment tool as it is only used in people with skin problems.
Also, the authors should explain why they chose DLQI in the current study rather than some more specific questionnaires such as VitiQoL and VIS-22. This also refers to my question regarding the Introduction section.
Author Response
Please see the attachment.

This manuscript is a resubmission of an earlier submission. The following is a list of the peer review reports and author responses from that submission.
Round 1
Reviewer 1 Report
Good attempt by the authors to document mental stress in vitiligo patients. However, this aspect has been established in many previous studies.
The authors have chosen to use less specific scores like DLQI, SF-36, WTP instead of specific scores like VIS-22, vITIQoLs etc. Besides non-availability of these scores in local language, was there any other reason not to use these scores in the present study?
How was the sample size calculated in the present study? Kindly explain whether the included numbers were sufficient to calculate the DLQI cutoff using ROC?
Why were cases with non-dermatological skin conditions like hypertension, DM not excluded? Around 26 out of 143 cases had these problems. Won't these conditions affect the overall impact on the QOL?
Mention the language in which the DLQI, SF-36 administered to the cases?
What was the education status of cases? Were there any illiterate cases, how was questionnaire administered to them?
How many had progressive vitiligo?
What was the exact primary and secondary objectives of the study? Kindly mention in the methodology.
Vitiligo does not cause physical symptoms like pain and itch. Kindly stress upon this aspect and discuss with findings of your study.
Reviewer 2 Report
comments for AA
Dear AA, the aim of your paper is very interesting, but our suggestion is to conform the text to scientific article guidelines, in particular concerning data of materials and methods, results and discussion. In the present form the paper is difficult to read and understand.
Line 31- Dear AA our suggested definition of vitiligo is “an immune-mediated skin condition resulting in a loss of pigmentation “. It is not only acquired but can also characterize some genetic pathologies. Please, would the AA better define the condition.
Lines 59-60 – would the AA explain what it means. We suggest AA simplifying the paragraph indicated above.
Lines 61-66 – we suggest the AA organize and expose the concept “In this study, we examined the impact of vitiligo via DLQI, 199 SF-36, and WTP assessments” without any other repetition. Then, we suggest you organize the introduction better, it sounds redundant, in particular not to repeat the same concept too much, indeed it confuses the reader.
Lines 72-73 – please would the AA specify only the characteristics of patients enrolled in the study.
Lines 95-96 – please the AA must only describe ranges of the scoring system without comments or personal evaluation.
Lines 127-133 – the described characteristics are not results but must be included in materials and methods.
Lines 145-146 – we suggest the AA report personal evaluations in the discussion paragraph.
Lines 241-245 – The aim to reassess DLQI scoring evaluation based on ROC analysis, we suggest not only removing from DLQI questionnaire, sections about systemic symptoms or pain, but adding some parameters or indicators suitable to enhance sensitivity of the test in vitiligo.
Round 2
Reviewer 2 Report
We suggest that for designing future studies, the authors use universally accepted standards of reference: the standard of reference of the present study was taken from a single article on rheumatoid arthritis (Usefulness of the SF-36 Health Survey in screening for depressive and anxiety disorders in rheumatoid arthritis. BMC Musculoskelet Disord. 2016;17:224. Published 2016 May 23. doi:10.1186/s12891-016-1083-y), instead of standards provided in the SF-36 Health Survey Manual & interpretation guide (Ware JE, Kosinski M, Gandek B: SF-36 Health Survey Manual & interpretation guide. 2004, Lincoln, RI, Qualiymetric Incorporated)We also advise that the authors express concepts more clearly and concisely, specifying what they are referring at in each context.
As final remark, a major language revision is advisable.